# Impact of sex and marital status on the prevalence of perceived depression in association with food insecurity

**Jung Woo Lee[1], Woo-Kyoung Shin[2], Yookyung Kim[1]***

**1** Department of Human Ecology, Graduate School, Korea University, Seoul, Republic of Korea,
**2** Department of Preventive Medicine, Seoul National University College of Medicine, Seoul, Republic of Korea

* yookyung_kim@korea.ac.kr

## Abstract

**Data Availability Statement:** Data can be freely downloaded without restriction from the website of the Korea Center for Chronic Disease and Control Institutional Data Access: https://knhanes.cdc.go.kr/knhanes/eng/index.do"

### Background

While both food insecurity and depression have been reported to be closely related to sex and marital status, the impact of sex and marital status on the prevalence of perceived depression in association with food security status has not been evaluated.

### Materials & methods

We performed a nationwide population study using data for 19,866 adults obtained from the 2012–2015 Korean National Health and Nutrition Examination Surveys. Household food insecurity status was evaluated using the 18-item Food Security Survey Module. Perceived depression was measured using one item questionnaire or the 9-item Patient Health Questionnaire (PHQ-9). We cross-sectionally analyzed associations between perceived depression and variables, including socio-demographic factors and food security status. The prevalence of perceived depression was compared according to sex, marital status, and food security status. We applied survey sampling weights in all analyses.

### Results

The overall prevalence of perceived depression was 10.5%. Prevalence rates of perceived depression in the high food security group, marginal food security group, low food security group, and very low food security group were 8.9%, 13.6%, 19.7%, and 35.0%, respectively ($P < 0.001$). Of total participants, 1.8% were categorized as having both perceived depression and food insecurity. After adjusting for confounding covariates, female sex (adjusted odds ratio [aOR]; 2.37), never married (aOR; 1.37), divorced/widowed/separated (aOR; 1.50), low food security (aOR; 1.72), and very low food security (aOR; 3.65) were associated with increased risk of perceived depression. Men with very low food security and divorced/widowed/separated status were most likely to have perceived depression (53.2%), followed by women with very low food security and divorced/widowed/separated status (48.7%),

**Funding:** The authors received no specific funding for this work.

**Competing interests:** The authors have declared that no competing interests exist.

women with very low food security and married status (42.0%), and women with low food security and divorced/widowed/separated status (33.3%).

## Conclusions

Female sex and marital status of divorced/widowed/separated were strongly associated with perceived depression. These two factors and food insecurity synergistically contributed to perceived depression.

## Introduction

Food insecurity is defined as limited access to food at the level of individuals or households due to lack of money or other resources [1]. Food insecurity components include insufficient food quantity, inadequate quality, unsafety, and cultural unacceptability [2]. Beyond hunger and increased risk of malnutrition, food insecurity is closely associated with a higher prevalence of chronic diseases such as diabetes mellitus, obesity, hypertension, hyperlipidemia, and metabolic syndrome [3, 4]. Globally, it has been estimated that nearly 821 million people remain food-insecure [1]. Even in the high-income region, more than 10% of all households are suffering from food insecurity [5, 6]. Food insecurity has been recognized as one of the key social determinants of health and a contemporarily important public health issue.

Food insecurity has been associated with unhealthy dietary patterns, including higher consumption of sugar/carbohydrate/meat/alcohol and lower consumption of fish/seafood [7]. Recent cross-sectional and longitudinal studies have shown that unhealthy dietary patterns can adversely affect psychological health [8–11]. Growing evidence has shown that food insecurity is closely linked to depression [12–15]. Both food insecurity and depression can negatively affect the lives of individuals. They can be strongly affected by socioeconomic factors, of which female sex and dissolution of marriage have been recognized as robust risk factors for food insecurity [16, 17] and depression [18–20]. However, to the best of our knowledge, no study has reported the impact of sex or marital status on the prevalence of depression across food insecurity categories. A better understanding of those associations could help us develop and evaluate strategies for economic and social support. Therefore, the objective of this study was to examine the impact of sex and marital status on the prevalence of perceived depression in association with food security status in Korean adults using nationwide population-based data.

## Materials and methods

### Design and study population

We used nationwide population-based cross-sectional data from the Korean National Health and Nutrition Examination Surveys (KNHANES). The KNHANES is a continuous, nationally representative survey conducted by the Korea Centers for Disease Control and Prevention (KCDC) [21]. KNHANES was designed to assess the health and nutrition status of the Korean people. It surveys non-institutionalized civilian Korean population. The KNHANES comprises in-person health interviews, health examinations, and a nutrition survey. We used 2012–2015 KNHANES data. Among 24,327 adult participants aged 19 years or older, we excluded participants whose household food security data (n = 1,901) or perceived depression data (n = 2,560) were missing. The final analytic sample consisted of 19,866 adults (Fig 1). The KNHNES was

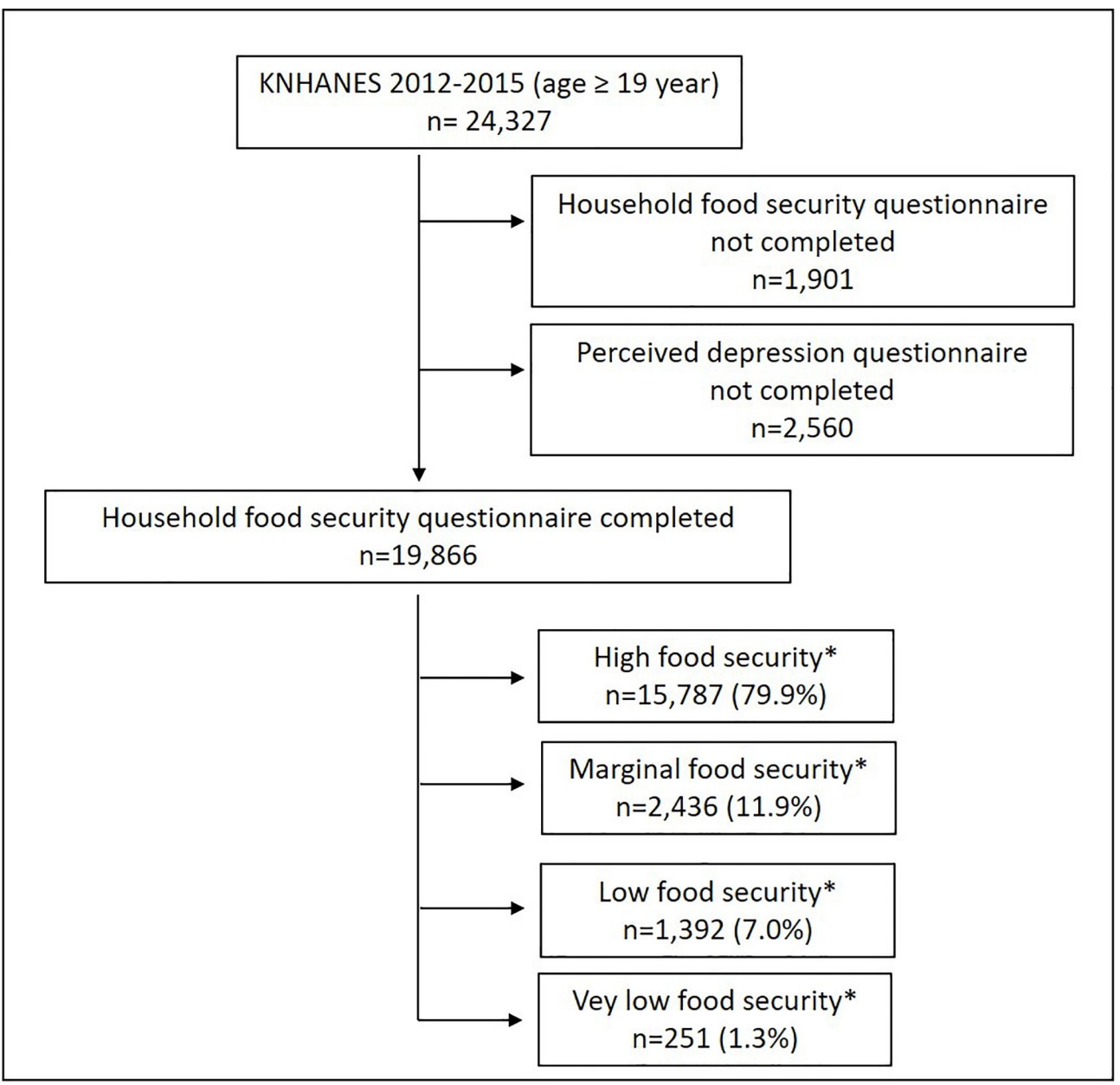

**Fig 1. Enrollment process for adults with reasons for exclusion.**

approved by the Ethics Committee of the KCDC. All participants provided informed consent. We used only publicly available data at http://knhanes.cdc.go.kr/knhanes.

## Measures

**Food security status.** Food security status was measured at the household level using an 18-item food insecurity questionnaire, which had been modified from the US Household Food Security Survey Module [22]. Responses to these items were scored. Food security was then

categorized as high food security, marginal food security, low food security, and very low food security. We considered households with low or very low food security as food-insecure.

**Perceived depression.** Perceived depression was assessed using the 9-item Patient Health Questionnaire (PHQ-9) (the year of 2014) or one-item questionnaire (the year of 2012, 2013, or 2015) which asked subject in a dichotomous manner (yes/no) on whether he or she felt sadness or despair enough to disturb daily life for more than 14 days consecutively over the past year. A dichotomous variable indicating no perceived depression (one-item questionnaire: no or PHQ-9 total score < 10) or perceived depression (one-item questionnaire: yes or PHQ-9 total score ≥ 10) [23] was created.

**Marital status.** Marital status was divided into five categories: never married, married, divorced, widowed, or separated. We reclassified these into three categories: never married, married (reference group), or divorced/widowed/separated.

**Other covariates.** Other sociodemographic covariates included age, education attainment, household income, smoking, alcohol intake, and physical activity. Age was classified into four categories: 19–39 years, 40–59 years (reference group), and more than 60 years. Education attainment was recoded into ≤elementary school, middle school, high school, and ≥ college education (reference group). Household income was divided into quartiles for lowest, lower-middle, upper-middle, or highest (reference group). Smoking status was categorized as current smoker, past smoker, and never smoker (reference group). Alcohol intake was categorized as heavy, moderate, and none (reference group). Those who drank at least seven glasses for men or 5 glasses for women at a time and more than twice per week were considered as heavy alcohol drinkers. Physical activity was divided into regular (reference group), intermittent, and none. Vigorous exercise more than four days per week was considered as having regular exercise. Anthropometric measurements were conducted by trained staff members. Body mass index (kg/m$^2$) was calculated as weight divided by the square of height.

## Statistical analyses

Data are presented as a percentage with a standard error for categorical variables and mean with a standard error for continuous variables. Unadjusted differences in socio-demographic characteristics across food security categories were tested using Rao-Scott Chi-square test or one-way analysis of variance (ANOVA). Multiple logistic regression was used to calculate the odds ratio and 95% confidence interval for perceived depression after adjusting for confounding covariates. Potential confounding covariates included sex, age, education attainment, marital status, income, smoking, alcohol, physical activity, food security status, and body mass index. We then compared the difference of perceived depression prevalence according to sex, marital status, and food insecurity status. Based on the statistical guidelines of the KCDC, we applied survey sampling weights in all analyses. We used SURVEY commands in SAS to account for the complex sampling strategy and produce output that was representative of the total Korean population. All statistical analyses were performed using SAS 9.4 software (SAS Institute, Cary, NC, USA). A *P* value of < 0.05 was considered statistically significant.

## Results

Fig 1 includes the distribution of food security status of finally enrolled 19,866 participants. Approximately half (49.3%) of these participants were males. The mean age of all participants was 45.8 years. Percentages (standard error) of those with high food security, marginal food security, low food security, and very low food security were 79.9% (0.6), 11.9% (0.4), 7.0% (0.3), 1.3% (0.1), respectively. Table 1 shows the characteristics of participants stratified by household food security status. Food-insecure participants (low food security or very low food

**Table 1. Characteristics of participants stratified by household food security status.[a].**

| Variable | Food secure n = 18,223 | | Food insecure n = 1,643 | | | |
| --- | --- | --- | --- | --- | --- | --- |
| | High food security n = 15,787 | Marginal food security n = 2,436 | Low food security n = 1,392 | Very low food security n = 251 | Total n = 19,866 | P value[b] |
| Sex | | | | | | < 0.001 |
| Male | 50.1 (0.4) | 47.5 (1.1) | 43.5 (1.6) | 45.6 (3.6) | 49.3 (0.4) | |
| Female | 50.0 (0.4) | 53.0 (1.1) | 56.5 (1.6) | 54.4 (3.6) | 50.7 (0.4) | |
| Age, mean years | 45.2 (0.2) | 48.6 (0.5) | 47.5 (0.6) | 49.1 (1.4) | 45.8 (0.2) | < 0.001 |
| 19–39 | 39.8 (0.7) | 33.9 (1.4) | 34.1 (1.7) | 29.9 (3.8) | 38.6 (0.6) | |
| 40–59 | 41.0 (0.6) | 36.1 (1.2) | 41.3 (1.7) | 42.9 (3.6) | 40.5 (0.5) | |
| > 60 | 19.2 (0.5) | 29.9 (1.3) | 24.6 (1.4) | 27.3 (3.4) | 20.9 (0.4) | |
| Marital status | | | | | | < 0.001 |
| Never married | 22.6 (0.6) | 23.5 (1.3) | 22.5 (1.6) | 22.8 (3.3) | 22.7 (0.5) | |
| Married | 69.4 (0.6) | 60.6 (1.4) | 54.4 (1.9) | 39.3 (4.1) | 66.9 (0.6) | |
| Divorced/separated/widowed | 8.0 (0.3) | 15.8 (0.9) | 23.1 (1.5) | 37.9 (4.0) | 10.3 (0.3) | |
| Education attainment | | | | | | < 0.001 |
| ≤ elementary school | 13.7 (0.4) | 25.6 (1.1) | 28.1 (1.5) | 39.0 (4.2) | 16.4 (0.4) | |
| Middle school | 8.3 (0.3) | 12.4 (0.9) | 11.7 (1.1) | 14.3 (2.9) | 9.1 (0.3) | |
| High school | 38.6 (0.6) | 40.0 (1.4) | 41.2 (1.8) | 31.2 (4.3) | 38.8 (0.5) | |
| ≥ College | 39.5 (0.7) | 22.0 (1.3) | 19. 0 (1.5) | 15.5 (3.3) | 35.7 (0.6) | |
| Income | | | | | | < 0.001 |
| 1st quartile (lowest) | 10.4 (0.4) | 27.8 (1.6) | 32.7 (2.0) | 58.3 (4.8) | 14.6 (0.5) | |
| 2nd quartile | 22.4 (0.6) | 34.1 (1.7) | 40.0 (2.3) | 27.9 (4.4) | 25.0 (0.6) | |
| 3rd quartile | 31.3 (0.7) | 24.3 (1.5) | 20.6 (1.8) | 12.7 (3.7) | 29.5 (0.6) | |
| 4th quartile (highest) | 35.9 (0.9) | 13.7 (1.3) | 6.7 (1.4) | 1.1 (1.1) | 30.8 (0.8) | |
| Smoking | | | | | | < 0.001 |
| current | 22.0 (0.5) | 25.2 (1.2) | 27.3 (1.6) | 31.5 (3.9) | 22.9 (0.4) | |
| past | 21.0 (0.4) | 17.9 (0.9) | 17.3 (1.2) | 16.8 (2.9) | 20.4 (0.3) | |
| none | 56.9 (0.5) | 56.8 (1.3) | 55.4 (1.6) | 51.7 (4.0) | 56.7 (0.4) | |
| Alcohol intake | | | | | | < 0.001 |
| Heavy | 19.6 (0.5) | 16.8 (1.1) | 16.3 (1.3) | 16.8 (3.6) | 19.0 (0.4) | |
| Moderate | 51.9 (0.6) | 46.5 (1.4) | 45.1 (1.8) | 33.5 (3.8) | 50.5 (0.5) | |
| None | 28.5 (0.5) | 36.7 (1.4) | 38.6 (1.7) | 49.7 (4.2) | 30.5 (0.5) | |
| Physical activity | | | | | | < 0.001 |
| None | 38.3 (0.6) | 42.4 (1.4) | 45.7 (1.8) | 51.8 (4.2) | 39.4 (0.5) | |
| Intermittent | 33.6 (0.5) | 32.1 (1.3) | 31.6 (1.7) | 28.9 (3.9) | 33.2 (0.5) | |
| Regular | 28.2 (0.5) | 25.5 (1.1) | 22.7 (1.4) | 19.3 (3.3) | 27.3 (0.4) | |
| Body mass index | | | | | | 0.01 |
| Underweight | 4.4 (0.2) | 4.6 (0.6) | 5.2 (0.7) | 6.6 (1.9) | 4.5 (0.7) | |
| Normal weight | 63.6 (0.5) | 61.8 (1.3) | 59.0 (1.6) | 56.7 (3.9) | 62.9 (0.4) | |
| Overweight | 32.0 (0.5) | 33.6 (1.3) | 35.8 (1.6) | 36.7 (3.7) | 32.5 (0.4) | |
| Perceived depression | | | | | | < 0.001 |
| No | 91.1 (0.3) | 86.4 (0.8) | 80.3 (1.4) | 65.0 (3.9) | 89.5 (0.3) | |
| yes | 8.9 (0.3) | 13.6 (0.8) | 19.7 (1.3) | 35.0 (3.9) | 10.5 (0.3) | |

[a]Data are presented as weighted percentage (standard error).

[b]P values for differences between food-secure participants and food-insecure participants.

**Table 2. Unadjusted and adjusted odds ratio and 95% confidence interval for perceived depression.**

|  | Unadjusted odds ratio (95% confidence interval) | *P* value | (95% confidence interval) | *P* value |
|---|---|---|---|---|
| Sex |  |  |  |  |
| Male | 1 |  | 1 |  |
| Female | 2.06 (1.48–2.86) | <0.001 | 2.39 (2.00–2.92) | < 0.001 |
| Marital status |  |  |  |  |
| Never married | 1.66 (1.16–2.39) | 0.01 | 1.37 (1.14–1.64) | < 0.001 |
| Married | 1 |  | 1 |  |
| Divorced/separated/widowed | 2.70 (2.03–3.60) | <0.001 | 1.48 (1.24–1.76) | < 0.001 |
| Food security status |  |  |  |  |
| High | 1 |  | 1 |  |
| Marginal | 2.46 (1.63–3.73) | <0.001 | 1.34 (1.12–1.60) | 0.001 |
| Low | 3.79 (2.41–5.93) | <0.001 | 1.75 (1.42–2.16) | < 0.001 |
| Very low | 8.24 (4.11–16.55) | <0.001 | 3.74 (2.62–5.33) | < 0.001 |
| Education |  |  |  |  |
| ≤ elementary school | 2.21 (1.51–3.23) | <0.001 | 1.71 (1.40–2.10) | < 0.001 |
| Middle school | 1.10 (0.65–1.87) | 0.72 | 1.82 (1.45–2.30) | < 0.001 |
| High school | 1.25 (0.87–1.80) | 0.23 | 1.27 (1.07–1.51) | 0.01 |
| ≥ College | 1 |  | 1 |  |
| Household income |  |  |  |  |
| 1st quartile (lowest) | 3.53 (2.25–5.53) | <0.001 | 1.33 (1.07–1.66) | 0.01 |
| 2nd quartile | 1.37 (0.88–2.13) | 0.17 | 0.96 (0.79–1.17) | 0.69 |
| 3rd quartile | 0.78 (0.50–1.25) | 0.33 | 0.93 (0.76–1.13) | 0.45 |
| 4th quartile (highest) | 1 |  | 1 |  |
| Smoking |  |  |  |  |
| current | 1.21 (0.84–1.74) | 0.30 | 1.72 (1.36–2.17) | < 0.001 |
| past | 0.57 (0.38–0.87) | 0.01 | 1.38 (1.12–1.71) | < 0.01 |
| none | 1 |  | 1 |  |
| Alcohol intake |  |  |  |  |
| Heavy | 0.84 (0.51–1.41) | 0.52 | 1.03 (0.84–1.27) | 0.76 |
| Moderate | 0.62 (0.45–0.85) | 0.001 | 0.86 (0.75–0.99) | 0.03 |
| None | 1 |  | 1 |  |

security groups) were more likely to be female, divorced/widowed/separated, current smoker, and non-heavy alcohol drinker. Food-insecure participants also had lower educational attainment, lower-income, and lower physical activity. The overall prevalence of perceived depression was 10.5%. A point estimate of the prevalence of perceived depression in the year 2014 was lower than that of the other three years combined (6.5% vs. 11.9%, $P < 0.001$). The difference in point estimates was 5.4% (95% confidence interval: 4.2–6.7%). Of total participants, 1.8% were categorized as having both perceived depression and food insecurity. Prevalence rates of perceived depression in the high food security group, marginal food security group, low food security group, and very low food security group were 8.9%, 13.6%, 19.7%, 35.0%, respectively ($P < 0.001$).

Table 2 shows the unadjusted and adjusted odds ratio (aOR) and 95% confidence interval for perceived depression. Multiple logistic regression showed that female sex (aOR: 2.39), never married (aOR: 1.37), divorced/widowed/separated (aOR: 1.48), low food security (aOR: 1.75), and very low food security (aOR: 3.74) were associated with a greater likelihood of experiencing perceived depression. Other significant factors included non-college education

attainment (aOR: 1.27–1.82), lowest household income (aOR: 1.33), current smoking (aOR: 1.72), and past smoking (aOR: 1.38). Age, body mass index, and physical activity were not significantly associated with perceived depression. A separate analysis of data in the year 2014 and the other years (2012, 2013, or 2015) also showed similar results (S1 Table).

The prevalence of perceived depression in association with sex, marital status, and food security status is shown in Table 3 and Fig 2. Men with very low food security and divorced/widowed/separated status were most likely to have perceived depression (53.2%), followed by women with very low food security and divorced/widowed/separated status (48.7%), married women with very low food security (42.0%), and women with low food security and divorced/widowed/separated status (33.3%). These findings were consistent throughout the study period (S2 Table).

## Discussion

We demonstrated that female sex and divorced/widowed/separated marital status independently had a strong impact on the prevalence of perceived depression in adults. When one or more of these factors showed a link with food insecurity, the likelihood of perceived depression was much greater. These findings indicate that sex, marital status, and food security status should be taken into account together as key factors for perceived depression.

We found strong dose-response pattern associations for food insecurity and perceived depression in adults. That is, the magnitude of the association was the strongest among those who had very low food security (35.0%), followed by those with low food security (19.7%) and marginal food security (13.6%). These findings are consistent with results of prior studies [24–27]. The strength of the current study was that we used large population-based data not confined to socio-economically vulnerable subgroups. Given the cross-sectional design, however, we could not provide any conclusion of a causal relationship between food insecurity and perceived depression. Several longitudinal studies have shown that food insecurity and depression are related in a bidirectional manner [15, 28, 29]. Since food insecurity is a modifiable factor, food insecurity interventions may yield benefits for the prevention, early detection, and management of depressive symptoms. Recently, a USA group has demonstrated that participation in a Supplemental Nutrition Assistance Program (SNAP) can significantly reduce psychological distress after six months of participation [30]. An evaluation of a poverty-alleviation program for the ultra-poor in Bangladesh has also shown that food insecurity is the most

**Table 3. Prevalence of perceived depression in association with sex-marital staus and food security status[a].**

|  | High food security | Marginal food security | Low food security | Very low food security | Total | P value |
|---|---|---|---|---|---|---|
| Male |  |  |  |  |  |  |
| never married (n = 1,495) | 8.8 (1.0) | 3.9 (1.5) | 19.1 (4.2) | 24.3 (9.7) | 9.0 (0.8) | < 0.001 |
| married (n = 6,313) | 5.5 (0.4) | 5.8 (1.0) | 8.1 (1.6) | 11.3 (5.2) | 5.7 (0.4) | 0.13 |
| divorced/widowed/separated (n = 497) | 12.8 (2.4) | 15.6 (5.0) | 22.8 (5.8) | 53.2 (10.9) | 17.6 (2.1) | < 0.001 |
| Subtotal (n = 8,305) | 6.7 (0.4) | 5.6 (0.8) | 13.1 (1.8) | 28.4 (5.6) | 7.2 (0.4) | <0.001 |
| Female |  |  |  |  |  |  |
| never married (n = 1,478) | 12.8 (1.1) | 17.8 (3.3) | 21.7 (4.9) | 17.4 (7.9) | 14.1 (1.0) | 0.07 |
| married (n = 7,835) | 9.7 (0.5) | 20.9 (1.7) | 20.6 (2.1) | 42.0 (7.2) | 11.8 (0.5) | < 0.001 |
| divorced/widowed/separated (n = 2,248) | 16.3 (1.2) | 22.4 (2.6) | 33.3 (3.8) | 48.7 (6.4) | 21.4 (1.1) | < 0.001 |
| Subtotal (n = 11,561) | 11.1 (0.4) | 20.8 (1.3) | 24.8 (1.9) | 40.5 (4.9) | 13.7 (0.4) | <0.001 |

[a]Data are presented as weighted percentage (standard error).

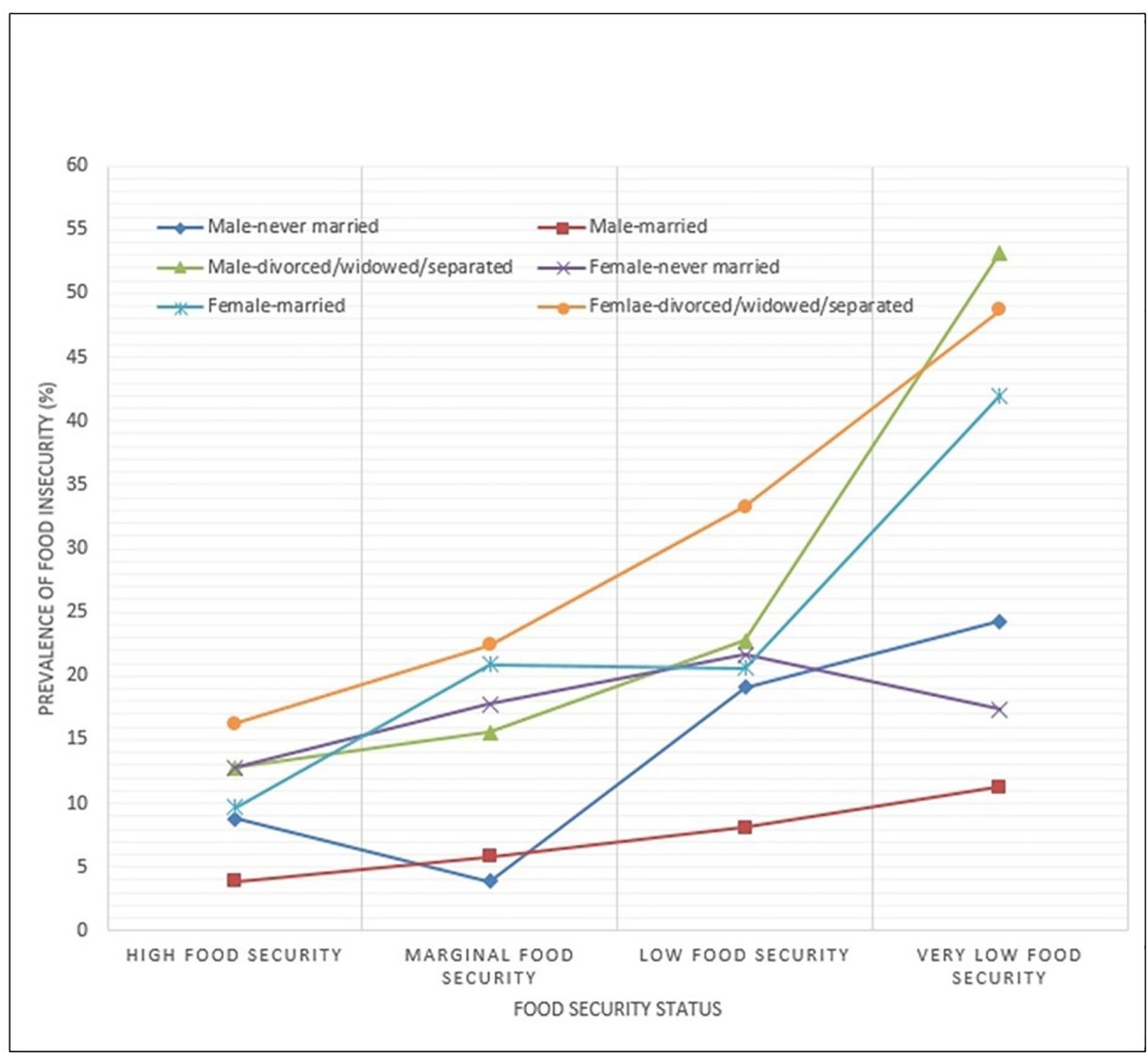

**Fig 2. Prevalence of perceived depression in association with sex, marital status, and food security status.**

important mediator for relieving psychosocial distress [31]. Further food insecurity interventional studies employing a study design relevant to regional socioeconomic context and population are warranted.

The novel aspect of this study is the investigation of the simultaneous effect of sex, marital staus, and food insecurity on perceived depression. While male sex was a strong protective factor for perceived depression, men with very low food security and divorced/widowed/separated status had the highest risk of perceived depression (53.2%). Even in the same group of men with very low food security, the rate of perceived depression widely ranged from 11.3% (married men) to 53.2% (divorced/widowed/separated men). Our findings indicate that concurrent analysis of significant factors for perceived depression and detailed subgroup analysis can be helpful for the determination of target population for support.

Our study has several limitations. First, as stated earlier, the cross-sectional study design precluded our ability to make a causal relationship. Second, we used a one-item questionnaire

in the year of 2012, 2013, or 2015 and PHQ-9 for the year of 2014. A point estimate of the prevalence of perceived depression in the year 2014 was lower than that of the other three years combined. Different usage of screening tool might have been associated with some biases. However, when we analyzed the data in the year 2014 and the other years, the main findings were similar regardless of the study period, supporting our conclusion. Third, the amount or impact of inadequate nutritional intake was not directly evaluated. Fourth, we grouped households with marginal food security as food secure group. There have been arguments that households with marginal food security have poorer adverse health outcomes than households with high food security [32]. Finally, we grouped subjects with divorced, widowed, or separated status into one group because of a relatively small number of each group of participants. Each status might have a differential impact on the perceived depression.

In conclusion, female sex and divorced/widowed/separated marital status were independent predictors for perceived depression in Korean adults. Food insecurity was closely associated with perceived depression in a dose-response fashion and synergistically contributed to a higher prevalence of perceived depression. These findings suggest that multidisciplinary efforts including economical, nutritional, and psychiatric support should be preferentially focused on these high-risk groups.

## Supporting information

**S1 Table. Adjusted odds ratio and 95% confidence interval for perceived depression.**
(DOC)

**S2 Table. Prevalence of perceived depression in association with sex-marital staus and food security status[a].**
(DOC)

## Author Contributions

**Conceptualization:** Jung Woo Lee, Woo-Kyoung Shin, Yookyung Kim.

**Data curation:** Jung Woo Lee, Woo-Kyoung Shin, Yookyung Kim.

**Formal analysis:** Jung Woo Lee, Woo-Kyoung Shin, Yookyung Kim.

**Writing – original draft:** Jung Woo Lee.

**Writing – review & editing:** Woo-Kyoung Shin, Yookyung Kim.

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
