## [Decision Letter · Decision Letter 0]

29 Apr 2020

PONE-D-19-21715

Impact of sex and marital status on the prevalence of perceived depression in association with food insecurity

PLOS ONE

Dear Yookyung　Kim

Thank you for submitting your manuscript to PLOS ONE. After careful consideration, we feel that it has merit but does not fully meet PLOS ONE’s publication criteria as it currently stands. Therefore, we invite you to submit a revised version of the manuscript that addresses the points raised during the review process.

We would appreciate receiving your revised manuscript by Jun 12 2020 11:59PM. To enhance the reproducibility of your results, we recommend that if applicable you deposit your laboratory protocols in protocols.io, where a protocol can be assigned its own identifier (DOI) such that it can be cited independently in the future. For instructions see: http://journals.plos.org/plosone/s/submission-guidelines#loc-laboratory-protocols

We look forward to receiving your revised manuscript.

Kind regards,

Yuka Kotozaki

Academic Editor

PLOS ONE

Reviewers' comments:

Reviewer's Responses to Questions

**Comments to the Author**

1. Is the manuscript technically sound, and do the data support the conclusions?

Reviewer #1: Yes

2. Has the statistical analysis been performed appropriately and rigorously? 

Reviewer #1: Yes

3. Have the authors made all data underlying the findings in their manuscript fully available?

Reviewer #1: Yes

4. Is the manuscript presented in an intelligible fashion and written in standard English?

Reviewer #1: Yes

5. Review Comments to the Author

Reviewer #1: This study is of interest and is mostly technically sound. The only concern is that the PHQ-9, a respectable screener for depressive symptoms, was only used in one year (2014). The other years used a yes/no response regarding symptoms for at least 14 days over the last year. At face value this is not totally unreasonable approach, but this reviewer would be like to know if the prevalence estimates for the PHQ-9 year (2014) were comparable to the other years (2012, 2013, 2015). For this purpose it will be sufficient to estimate the overall prevalence of depression in year 2014 vs the other 3 years combined, with a t test. 95% CIs would be preferable for these estimates as a more informative epidemiological approach than standard errors re the precision of the point estimates.

Re Table 1. BMI is not mentioned in the text unless I missed it

Model of Table 2: suggest dropping non-significant variables (age, BMI and physical activity). This will modify the other ORs a little, and that they are non-significant can be mentioned in the text.

Line 165: I do not understand the sentence beginning with “An alarmingly high…..”

6. PLOS authors have the option to publish the peer review history of their article (what does this mean?). If published, this will include your full peer review and any attached files.

Reviewer #1: No

---

## [Author Response · Author response to Decision Letter 0]

16 May 2020

Response letter

Dear Dr. Kotozaki,

Thank you for reviewing our manuscript PONE-D-19-21715 entitled “mpact of sex and marital status on the prevalence of perceived depression in association with food insecurity”. We appreciate your in-depth review and helpful comments. We have made corrections and modifications accordingly. The following are itemized lists stating our disposition in detail of each point raised by the reviewer. Thank you for your consideration of our manuscript. We look forward to a favorable decision on this revision.

Sincerely yours,

Yookyung Kim, Ph D

Professor 

Department of Human Ecology, Graduate School, Korea University 

#1 reviewer:

Reviewer comment: This study is of interest and is mostly technically sound. The only concern is that the PHQ-9, a respectable screener for depressive symptoms, was only used in one year (2014). The other years used a yes/no response regarding symptoms for at least 14 days over the last year. At face value this is not totally unreasonable approach, but this reviewer would be like to know if the prevalence estimates for the PHQ-9 year (2014) were comparable to the other years (2012, 2013, 2015). For this purpose it will be sufficient to estimate the overall prevalence of depression in year 2014 vs the other 3 years combined, with a t test. 95% CIs would be preferable for these estimates as a more informative epidemiological approach than standard errors re the precision of the point estimates.

Response: Thanks for the reviewer’s thoughtful comments. As the reviewer recommended, we compared the overall prevalence of depression in the year 2014 vs. the other three years combined. To see whether different usage of screening tool has influenced our conclusion, we also separately performed multiple regression analysis for perceived depression in the year 2014 vs. the other three years. A t-test showed that the point estimate of the prevalence of perceived depression in the year 2014 was lower than that of 3 years combined (5.6% vs. 11.9%, P < 0.001). The difference in point estimates was 5.4% (95% confidence interval: 4.2-6.7%). Multiple regression analysis showed similar results regardless of the year/period. That is, female sex and dissolution of marriage had a significant impact prevalence of perceived depression and synergistically contributed to a higher prevalence of perceived depression with food insecurity. We described these findings in the text (page 7, line 119-122; page 8, line 133-134. line 140-141; page 10, line 176-180 ) and newly prepared Supplementary Tables 1 & 2. 

Reviewer comments: Re Table 1. BMI is not mentioned in the text unless I missed it. Model of Table 2: suggest dropping non-significant variables (age, BMI and physical activity). This will modify the other ORs a little, and that they are non-significant can be mentioned in the text.

Response: As the reviewer recommended, we deleted non-significant variables in Table 2. We mentioned it in the text (page 8, line 132-134). We revised the ORs in Table 2 (page 15: Adjusted odds ratio) and the text (page 8, line 127-131). 

Reviewer comment: Line 165: I do not understand the sentence beginning with “An alarmingly high…..”

Response: We deleted the sentence in the revised manuscript (page 8, line 168-169).

---

## [Decision Letter · Decision Letter 1]

20 May 2020

Impact of sex and marital status on the prevalence of perceived depression in association with food insecurity

PONE-D-19-21715R1

Dear Dr. Yookyung Kim,

We are pleased to inform you that your manuscript has been judged scientifically suitable for publication and will be formally accepted for publication once it complies with all outstanding technical requirements.

With kind regards,

Yuka Kotozaki

Academic Editor

PLOS ONE

Additional Editor Comments (optional):

Reviewers' comments:

Reviewer's Responses to Questions

**Comments to the Author**

1. If the authors have adequately addressed your comments raised in a previous round of review and you feel that this manuscript is now acceptable for publication, you may indicate that here to bypass the “Comments to the Author” section, enter your conflict of interest statement in the “Confidential to Editor” section, and submit your "Accept" recommendation.

Reviewer #1: All comments have been addressed

2. Is the manuscript technically sound, and do the data support the conclusions?

Reviewer #1: (No Response)

3. Has the statistical analysis been performed appropriately and rigorously? 

Reviewer #1: (No Response)

4. Have the authors made all data underlying the findings in their manuscript fully available?

Reviewer #1: (No Response)

5. Is the manuscript presented in an intelligible fashion and written in standard English?

Reviewer #1: (No Response)

6. Review Comments to the Author

Reviewer #1: (No Response)

7. PLOS authors have the option to publish the peer review history of their article (what does this mean?). If published, this will include your full peer review and any attached files.

Reviewer #1: Yes: Dr Andrew Bulloch

---

## [Editor Report · Acceptance letter]

1 Jun 2020

PONE-D-19-21715R1 

Impact of sex and marital status on the prevalence of perceived depression in association with food insecurity 

Dear Dr. Kim:

I am pleased to inform you that your manuscript has been deemed suitable for publication in PLOS ONE. Congratulations! Your manuscript is now with our production department. 

With kind regards,

on behalf of

Dr. Yuka Kotozaki 

Academic Editor

PLOS ONE